# Prediction of the Risk of Malignancy of Adnexal Masses during Pregnancy Comparing Subjective Assessment and Non-Contrast MRI Score (NCMS) in Radiologists with Different Expertise

**DOI:** 10.3390/cancers15215138

**Published:** 2023-10-25

**Authors:** Camilla Panico, Silvia Bottazzi, Luca Russo, Giacomo Avesani, Veronica Celli, Luca D’Erme, Alessia Cipriani, Floriana Mascilini, Anna Fagotti, Giovanni Scambia, Evis Sala, Benedetta Gui

**Affiliations:** 1Department of Diagnostic Imaging, Oncological Radiotherapy and Haematology, Fondazione Policlinico Universitario A. Gemelli IRCCS, 00168 Rome, Italy; camilla.panico@policlinicogemelli.it (C.P.); silvia.bottazzi01@icatt.it (S.B.); luca.russo@guest.policlinicogemelli.it (L.R.); veronica.celli@guest.policlinicogemelli.it (V.C.); luca.derme01@icatt.it (L.D.); alessia.cipriani03@icatt.it (A.C.); evis.sala@policlinicogemelli.it (E.S.); benedetta.gui@policlinicogemelli.it (B.G.); 2Department of Woman and Child Health and Public Health, Fondazione Policlinico Universitario A. Gemelli IRCCS, 00168 Rome, Italy; floriana.mascilini@policlinicogemelli.it (F.M.); anna.fagotti@policlinicogemelli.it (A.F.); giovanni.scambia@policlinicogemelli.it (G.S.); 3Faculty of Medicine and Surgery, Università Cattolica del Sacro Cuore, 00168 Rome, Italy

**Keywords:** ovarian cancer, MRI, pregnancy

## Abstract

**Simple Summary:**

Characterising an ovarian mass during pregnancy is essential to avoid unnecessary treatment and, if treatment is required, to plan it accordingly. MRI of the pelvis with post-contrast sequences is indicated when adnexal masses are indeterminate at the US examination. However, the administration of intravenous gadolinium-based contrast agents is a method that should have a limited use in pregnant women. We evaluated the diagnostic accuracy of the Non-Contrast MRI Score (NCMS) in pregnant women, using both a subjective assessment (SA) and the NCMS, between two radiologists with different expertise. Relying on histopathology and imaging follow-up at one year as the gold standard, we found that the expert radiologist correctly classified 90% of the diagnoses using both SA and the NCMS (85.7% sensitivity and 92.3% specificity, with a false positive rate of 7.7% and a false negative rate of 14.3%). The non-expert radiologist correctly identified patients at a lower rate, especially using the SA (60%), with a sensitivity of 85% and a specificity of 46.2%. The analysis of the inter-observer agreement showed a K = 0.47 (95% CI: 0.48–0.94) for the SA (agreement in 71.4% of cases) and a K = 0.8 (95% CI: 0.77–1.00) for the use of the NCMS (agreement in 90% of cases). This study evaluates the diagnostic accuracy of non-contrast MRI scores in pregnant women with indeterminate ovarian masses at the US examination. The NCMS is a reliable tool to predict the risk of malignancy of adnexal masses in these women and extremely useful for inexperienced radiologists.

**Abstract:**

Ovarian cancer represents 7% of all cancers in pregnant women. Characterising an ovarian mass during pregnancy is essential to avoid unnecessary treatment and, if treatment is required, to plan it accordingly. Although ultrasonography (US) is the first-line modality to characterise adnexal masses, MRI is indicated when adnexal masses are indeterminate at the US examination. An MRI risk stratification system has been proposed to assign a malignancy probability based on the adnexal lesion’s MRI, but features of the scoring system require the administration of intravenous gadolinium-based contrast agents, a method that might have a limited use in pregnant women. The non-contrast MRI score (NCMS) has been used and evaluated in non-pregnant women to characterise adnexal masses indeterminate at the US examination. Therefore, we evaluated the diagnostic accuracy of the NCMS in pregnant women, analysing 20 cases referred to our specialised institution. We also evaluated the diagnostic agreement between two radiologists with different expertise. The two readers classified ovarian masses as benign or malignant using both subjective assessment (SA), based on the interpretive evaluation of imaging findings derived from personal experience, and the NCMS, which includes five categories where 4 and 5 indicate a high probability of a malignant mass. The expert radiologist correctly classified 90% of the diagnoses, using both SA and the NCMS, relying on a sensitivity of 85.7% and a specificity of 92.3%, with a false positive rate of 7.7% and a false negative rate of 14.3%. The non-expert radiologist correctly identified patients at a lower rate, especially using the SA. The analysis of the inter-observer agreement showed a K = 0.47 (95% CI: 0.48–0.94) for the SA (agreement in 71.4% of cases) and a K = 0.8 (95% CI: 0.77–1.00) for the NCMS (agreement in 90% of cases). Although in pregnant patients, non-contrast MRI is used, our results support the use of a quantitative score, i.e., the NCMS, as an accurate tool. This procedure may help less experienced radiologists to reduce the rate of false negatives or positives, especially in centres not specialised in gynaecological imaging, making the MRI interpretation easier and more accurate for radiologists who are not experts in the field, either.

## 1. Introduction

The incidence of cancer during childbearing is rising due to the increasing age of first pregnancy [1]. Ovarian cancer represents 7% of all cancers in pregnant women [2]. The frequency of adnexal masses in pregnancy differs between different studies—approximately between 0.05% and 2.4% of all pregnancies [3,4,5,6]. Most of the masses are benign [3,4,5,6], and malignant ones are reported between 1 and 8% [3,4,5,6]. Therefore, characterising an ovarian mass during pregnancy is essential to avoid unnecessary treatment and, if treatment is required, to plan it accordingly.

Ultrasound (US) is the first-line modality to characterise adnexal masses, and various diagnostic scores have been proposed and evaluated for sensitivity and specificity to diagnose malignant lesions; the IOTA’s (International Ovarian Tumour Analysis) simple rules provide a sensitivity of 92% and a specificity of 69% [6,7]. MRI is indicated when adnexal masses are indeterminate at the US examination [8,9,10]. The success of the use of MRI was due to its better detection of fat and blood content in lesions.

High rates of MRI were reported in young women (pregnant and non-pregnant) for diagnoses in many body organs. Regarding MRI in pregnancy, the prevalence of 1 gadolinium exposure every 860 pregnancies was detected in a large American study [11]. This prevalence was 4.3-fold greater during the first trimester than during the second trimester and 5.1-fold greater than during the third trimester. In non-pregnant women, the Ovarian–Adnexal Imaging–Reporting–Data System (O-RADS) MRI risk stratification system has been proposed to assign a malignancy probability based on the adnexal lesion’s MRI features [12,13]. The score yielded high sensitivity (93%) and specificity (91%) for stratifying the risk of malignancy in adnexal masses [13]. Moreover, Thomassin-Naggara et al. showed that MRI has a high accuracy (>80%) in the characterisation of adnexal masses in pregnancy, especially when evaluating images using the AdnexMR score, in a cohort study of 1340 women [12]. Both these scoring systems require the administration of intravenous gadolinium-based contrast agents (GBCAs) [12,13]. Some clinical studies have shown that, despite gadolinium-based contrast media during pregnancy being relatively safe, with a low incidence of adverse effects, its use should be limited in clinical practice when strictly necessary [14,15,16,17]. Recently, the non-Contrast MRI score (NCMS) was proposed [18]. The score includes five categories—from 1 to 5—and does not require the administration of GBCAs (Table 1).

The use of the NCMS, which has already shown promising results in non-pregnant patients, may have an added value in the clinical scenario of pregnancy when the use GBCAs is discouraged.

This study aims to investigate the accuracy of non-contrast MRI in the prediction of the risk of malignancy of indeterminate adnexal masses at ultrasonography in pregnancy, comparing subjective assessments (SAs) and NCMSs between radiologists with “different expertise”.

Considering the wide range of possible diagnoses, an additional task of the radiologist was to provide diagnostic hypotheses regarding the nature of the mass.

## 2. Materials and Methods

### 2.1. Study Design

This is a retrospective monocentric study. 

The institutional ethics committee approved this study (approval no. 5681, 5 June 2023).

Inclusion criteria were the following: (1) pregnant women; (2) >18 years old; (3) indeterminate adnexal masses at US examination; (4) availability of MRI and histopathological results or at least one follow-up imaging at one year. Patients with a diagnosis of extra-ovarian mass were excluded.

Our picture archiving and communication system (PACS) was searched to retrieve the MRI examinations of pregnant patients scanned for a US-indeterminate adnexal mass between January 2011 and February 2023. All patients meeting the inclusion criteria were asked to sign informed consent forms to use their clinical data. Only those who gave informed consent were included in the study.

### 2.2. MRI Protocol

Patients were scanned on different 1.5T MRI scanners from the same vendor (GE Medical Systems, Milwaukee, WI, USA), using a phased-array abdominal coil. The acquisition protocol is reported in Appendix A.

A gadolinium-based contrast agent was not administered. T2WI fast-spin echo (FSE) sequences were obtained in multiple planes (axial, sagittal and coronal). Axial T1WI gradient-echo or FSE sequences were obtained with and without fat suppression. Axial diffusion-weighted imaging (DWI) was acquired with b values of 0 and 800–1000 s/mm^2^. Apparent diffusion coefficient (ADC) maps were derived. In addition, axial T2WIs and axial DWIs of the upper abdomen were acquired to assess the presence of retroperitoneal lymphadenopathy and hydronephrosis. 

### 2.3. Image Interpretation

Two radiologists (reader 1 and reader 2) with different expertise in gynaecological imaging—1 and 7 years, respectively—independently reviewed the images blinded to US examination results. The two readers classified ovarian masses using SA and the NCMS. The lesions were classified according to the SA as benign or malignant. Subjective assessment was based on the interpretive evaluation of imaging findings based on personal experience, expertise and judgment. Then, the readers assigned a score from 1 to 5 following the NCMS (Table 1). The scoring system includes five categories. A score of 1 was assigned in cases where no adnexal masses were found. If a radiological diagnosis of a benign/likely benign mass was made (endometrioma, fibroma, or dermoid), the readers assigned a score of 2. Higher scores (3, 4 and 5) were assigned to masses characterised as indeterminate, suspicious and highly suspicious for malignancy, respectively. If solid tissue was observed, the lesions were scored as 4. Solid tissue was defined as tissue with intermediate signal intensity (SI) on the T2WI, low signal intensity on the T1WI, and diffusion restriction (high SI on the DWI and low SI on ADC maps). A score of 5 was assigned when lymphadenopathy, peritoneal implants, and/or ascites were present. If masses could not be classified in other scores, a score of 3 was given. Masses with an NCMS > 4 and adnexal lesions classified as indeterminate or malignant at the subjective assessment were considered malignant. Masses with an NCMS < 4 and classified as benign at the subjective assessment were considered benign.

### 2.4. Statistical Analysis

The categorical variables were expressed as absolute numbers and percentages. According to their distribution, continuous variables were described either as a median and interquartile range or as a mean and standard deviation. Sensitivity, specificity, positive predictive value (PPV), negative predictive value (NPV), accuracy, false positive rate and false negative rate were calculated for both the SAs and the NCMSs of reader 1 and reader 2. These values were calculated using the final diagnoses derived from histopathology (when available) or the follow-up imaging results as binary variables (benign or malignant). Borderline diagnoses were considered malignant. The values were presented as percentages (%) and a 95% confidence interval (CI). The inter-observer agreement between reader 1 and reader 2 for NCMSs and SAs was investigated using Cohen’s kappa statistic for agreement. Statistical analysis was performed using SPSS software (SPSS Statistics for Mac 24.0, IBM, SPSS Inc., Chicago, IL, USA) [19].

## 3. Results

Twenty pregnant women were included in this study. Table 2 reports their mean age and range, separated by diagnosis. Information about laterality, dimension and histologic results is included in Appendix A.

There was no difference in the diagnostic outcomes performed by the expert radiologist using either a subjective assessment (Table 3a) or the NCMS (Table 3b).

The diagnostic accuracy of the expert radiologist, reported in Table 3c, was the same for the use of SAs and the NCMS, with the following values: sensitivity, 85.7%; specificity, 92.3%; positive predictive value, 85.7%; negative predictive value, 92.3%; false positive rate, 7.7%; false negative rate, 14.3%. The expert radiologist correctly classified 90.0% of cases, using both SA and the NCMS.

Different results are shown for the non-expert radiologist with a low rate of correctly identified patients (12/20; 60%), especially for the SA (Table 4a,b), where the specificity was very low (46.2%).

Table 5a shows the inter-observer agreement between the non-expert and expert radiologists regarding the diagnostic outcomes with SAs: they agreed in 71.4% of the cases with a K = 0.47 (95% CI: 0.48–0.94). Table 5b shows the inter-observer agreement between the non-expert and the expert radiologists regarding the diagnostic outcomes with the use of the NMCS: they agreed in 90.5% of cases with a K = 0.8 (95% CI: 0.77–1.00).

The analytical description of the diagnoses performed by the two radiologists using the NCMS, according to each single score, is reported in Table 6. There were differences for score 3 (6/20 for the non-expert and 3/20 and for the expert) and for the diagnosis of malignant masses (more scores of 4 and 5 by the non-expert). It is interesting to note that the non-expert radiologist tended to overcall on the side of malignancy, with two benign masses classified by the expert radiologist as benign and as malignant by the non-expert one.

## 4. Discussion

This retrospective study aimed at investigating the accuracy of non-contrast MRI in characterising sonographically indeterminate adnexal masses upon US examination in pregnant women and also in evaluating the diagnostic accuracy of radiologists with different expertise. In our study, the diagnostic accuracy was high using the NCMS (80% of correctly classified diagnoses for the non-expert radiologist and 90% for the expert one). Furthermore, we found that the agreement between the expert and the non-expert radiologists using the SA was low, while it improved with the use of the NCMS, although the non-expert radiologist tended to over-diagnose malignancy.

Sahin et al. used a similar approach to characterise 350 adnexal masses in non-pregnant women, with promising results. An NCMS ≥4 was associated with malignancy with an accuracy of 94.2%, a sensitivity of 84.9%, a specificity of 95.9% and a very good positive likelihood ratio [18], when compared to histology or imaging follow-up at one year. These findings were comparable to those from contrast-enhanced studies [11]. The interpretation of images without post-contrast enhancement performed in our study was corroborated by the Sahin study.

Although exposure to MRI in pregnancy is relatively safe for the foetus [20,21], we know that GBCAs pass through the placental barrier and enter foetal circulation, increasing the risk of rheumatological diseases, inflammatory skin diseases and stillbirth or neonatal undesired birth outcomes [14,15,20,21], therefore restricting the use of contrast-enhanced MRIs to very select cases where the benefit outweighs the risk, only if the imaging is essential and cannot be delayed [20,22].

In this context, our study becomes more valuable.

In the literature, only one study characterised sonographically indeterminate adnexal masses in pregnant women with contrast-enhanced MRIs [23]. In this study, 88.9% of cases were correctly classified, and the result was very similar to our expert radiologist’s performance (90%) using the NCMS; this finding suggests that contrast is probably not necessary in this delicate scenario. A few other studies have reported data on the prevalence of malignant adnexal masses in pregnancy but not on the diagnostic accuracy of MRIs. Moreover, they are not comparable since they have been carried out in different settings, observing patients with different characteristics, and are not limited to ovarian masses undetermined at the US examination [3,24,25].

Our results highlighted that the diagnostic accuracy was affected by the fact that, in rare cases, there may be subtle differences between a benign and a malignant lesion, causing misinterpretations, thus leading to an increase in false negatives or false positives for both the expert and the non-expert radiologists. One example in our study population, also reported in the literature, was the correct interpretation of decidualised endometrioma [26,27,28,29,30,31,32]. During pregnancy, decidual endometrium changes are due to high levels of progesterone, so endometrioma can undergo decidualisation. Even if the decidualisation of ovarian endometrioma is rare, when it occurs, it is difficult to differentiate from ovarian cancer. In fact, the presence of irregular wall thickening in the decidualised endometrioma can mimic the papillary projections of ovarian cancer [26,32]. Supported by the existing literature, our results also suggested that the evaluation of the ADC value is key for the correct diagnosis because it is significantly higher in benign disease than in ovarian cancers [33] (Figure 1 and Figure 2). Another difficult case was the interpretation of a borderline cystadenoma, which was classified as benign using the SA, with a score of three in the NCMS by both the expert and non-expert radiologists due to the presence of multiple thin septations without the detection of macroscopic solid tissue. However, at histopathology, it was classified as malignant. No solid tissue was seen on the MRI images, due to the difficulty of detecting micro-invasive components [34] (Figure 3). On the other hand, the diagnosis of a benign mature cystic teratoma was correctly made by the expert radiologist, while the lesion was misclassified as malignant by the non-expert radiologist (NCMS 5). In fact, the fat component, which is the main sign of a dermoid cyst, was relatively minimal and was not identified by the non-expert radiologist (Figure 4).

Figure 5 and Figure 6 show two cases of correct diagnoses made by the expert and the non-expert radiologists: one case of a struma ovarii, a multilocular cystic mass without solid tissue that was correctly considered benign/NCMS 2 (true negative), and a case of a low-grade ovarian serous cancer with papillary projections and mural nodules (solid tissue) correctly classified as malignant/score 4 (true positive).

The strength of our study was the use of the histopathological results after surgery or the imaging follow-up after one year as the gold standard. Moreover, we evaluated patients with a standard multidisciplinary approach.

Our study has some limitations. First, the retrospective design. Second, the fact that it was performed in a highly specialised institution, and this may provide results that are not easily generalisable to other health institutions. Third, the low number of patients; however, our numbers are not very far from the 31 cases of the only other previous study investigating the diagnostic performance of MRI in pregnant women [23]. Moreover, adnexal masses in pregnancy are rare, and this is the reason for the low number of cases included; multicentric studies may be considered to improve the sample number.

Although in pregnant women, non-contrast MRI is applied, our results support the use of a quantitative score, i.e., the NCMS, as an accurate tool for the characterisation of adnexal masses. This procedure may help less experienced radiologists to reduce the rate of false negatives or positives, especially in centres not specialised in gynaecological imaging, making MRI interpretation easier and more accurate, also for radiologists who are not experts in the field. Nevertheless, it is crucial to emphasise that the patient’s overall clinical scenario has a pivotal role; clinical reasoning is the key to the diagnostic process and, together with imaging features, is fundamental for the radiologist to reach an accurate diagnosis.

## 5. Conclusions

Our study is the first one to evaluate the diagnostic accuracy of NCMSs in pregnant women with indeterminate ovarian masses upon US examination. We found that the NCMS is a reliable tool to predict the risk of malignancy of adnexal masses in pregnant patients. Moreover, the use of the NCMS is particularly useful for less experienced radiologists. Larger, multi-centre prospective studies are necessary to confirm and validate our results.

## Figures and Tables

**Figure 1 cancers-15-05138-f001:**
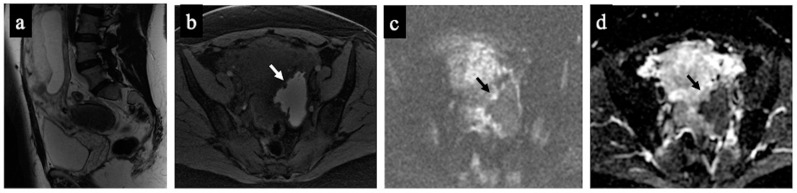
Decidualised endometrioma. MR images of a 37-year-old woman—16 weeks pregnant—with an indeterminate adnexal lesion discovered at the first-trimester US. The sagittal (**a**), axial T1-WI with fat-saturation (**b**), DWI (**c**) and ADC map (**d**) show a unilocular cystic left adnexal mass with haemorrhagic content (high signal intensity on the T1-WI with fat saturation) and some small papillary projections (arrows). Note how the small papillary projections have an intermediate signal on the T2-WI but no corresponding true diffusion restriction (high signal in both DWIs on ADC-map), so they are not considered “solid tissue” according to the NCMS. The lesion was considered indeterminate according to the SA by the non-expert radiologist, who correctly reclassified it as a score of 2 using the NCMS (false positive/true negative).

**Figure 2 cancers-15-05138-f002:**
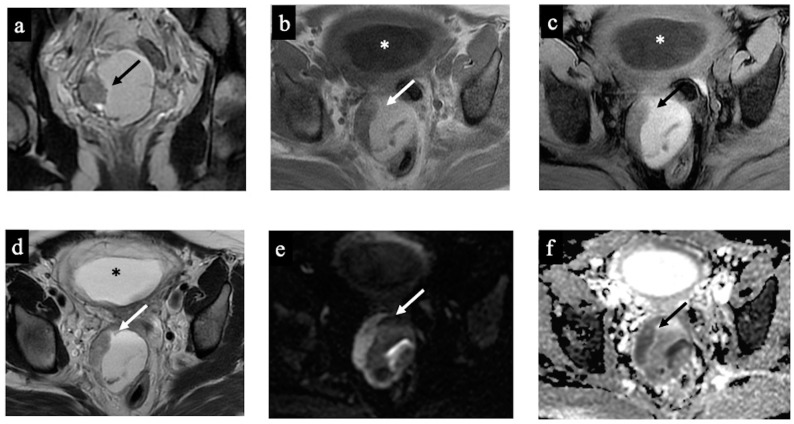
Decidualised endometrioma. MR images of a 34-year-old woman—20 weeks pregnant—with an indeterminate adnexal lesion discovered at the 16-week US. The coronal T2-WI (**a**), axial T1 (**b**), T1-WI with fat-saturation (**c**), T2-WI (**d**), DWI (**e**) and ADC-map images (**f**) show a unilocular cystic right adnexal mass with haemorrhagic content (high signal intensity on the T1-WI, T2-WI, and T1 with fat saturation). This was considered a false positive since both readers misclassified the mass as malignant/score 4 due to the presence of tissue with true diffusion restriction along the lesion’s right lateral wall (arrow). Note the gestational sac in the uterine cavity (asterisk).

**Figure 3 cancers-15-05138-f003:**
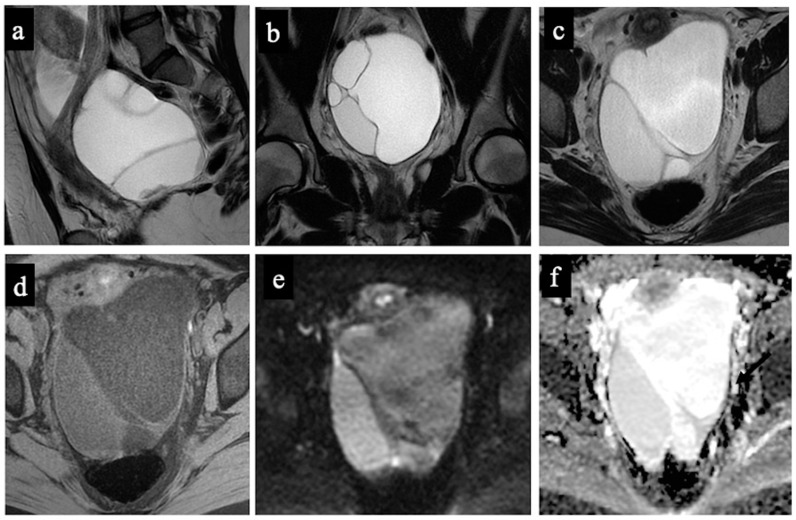
Borderline mucinous cystadenoma. MR images of a 28-year-old woman—17 weeks pregnant—with an indeterminate adnexal lesion discovered at the first-trimester US. The sagittal (**a**), coronal (**b**), and axial T2-WIs (**c**), T1-WI with fat-saturation (**d**), DWI (**e**) and ADC-map images (**f**) show a multilocular cystic left adnexal mass with different signal intensities within the loculi. Both readers considered this lesion as benign/score 3 (probable mucinous cystadenoma) because no solid tissue was found (false negative).

**Figure 4 cancers-15-05138-f004:**
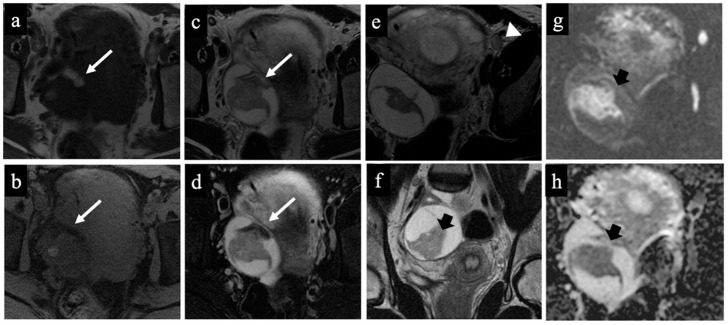
Dermoid cyst. MR images of a 32-year-old woman with an indeterminate adnexal lesion discovered at the first-trimester US. The axial T1-WI (**a**), T1-WI with fat-saturation (**b**), T2-WI (**c**,**e**), T2-WI with fat-saturation (**d**), coronal T2-WI (**f**), axial DWI (**g**) and ADC-map images (**f**) show a complex right adnexal lesion with fluid and fatty content. Note the drop of the signal of the fatty component, comparing images (**a**)/(**b**) and (**c**)/(**d**) (long arrow). The expert radiologist correctly classified the mass as benign/score 2 (true negative). The subtle fatty content, the left external iliac node with a short axis of 10 mm (arrowhead in image (**e**)), and the presence of tissue with true diffusion restriction within the mass (short arrow in images (**f**–**h**)), tricked the non-expert radiologist who classified it as malignant/score 5 (false positive).

**Figure 5 cancers-15-05138-f005:**
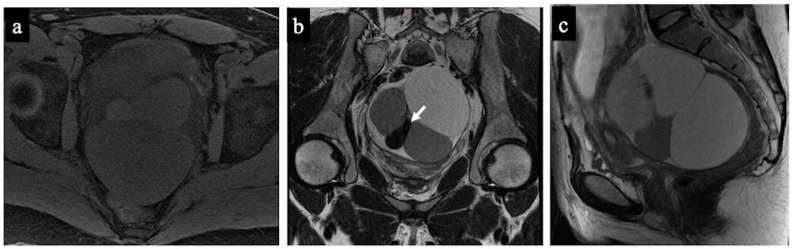
Struma Ovarii. MR images of a 34-year-old woman—16 weeks pregnant—with an indeterminate adnexal mass accidentally discovered at first trimester US. The axial T1 with fat saturation (**a**) and coronal (**b**) and sagittal T2-WI (**c**) images show a multilocular cystic mass with different signal intensities within the loculi. Some loculi have a very low signal intensity on the T2-WI (white arrow in image (**b**)) corresponding to colloid. No solid tissue is seen within the lesion; thus, it was correctly considered as benign/NCMS 2 according to both readers (true negative).

**Figure 6 cancers-15-05138-f006:**
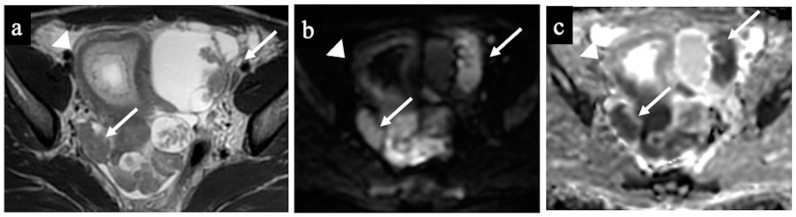
Low-grade ovarian serous cancer. MR images of a 33-year-old woman—7 weeks pregnant—with bilateral indeterminate adnexal lesions discovered at the 6-week US. The axial T2-WI (**a**), DWI (**b**) and ADC-map (**c**) show bilateral adnexal masses with papillary projections and mural nodules (solid tissue) within the lesions (arrows). Solid tissue has an intermediate signal on T2-WIs and corresponding true diffusion restriction. Free pelvic fluid is seen in the pouch of Douglas. No carcinosis was present, so the lesions were correctly classified as malignant/score 4 by both the readers (true positive). Note the gestational sac in the uterine cavity (arrowhead).

**Table 1 cancers-15-05138-t001:** The non-contrast MRI score (NCMS).

Score	Definition	MRI Features
1	No adnexal mass	No adnexal mass present
2	Benign/likely benign	Radiological diagnosis of benign mass (e.g., endometrioma, dermoid, fibroma)
3	Indeterminate	Not classified in another score (it may have a solid component, but it does not reach the criteria for solid tissue *)
4	Suspicious for malignancy	Solid tissue criteria reached *
5	Highly suspicious for malignancy	Solid tissue criteria reached * + lymphadenopathy/peritoneal implants/ascites

* Solid tissue is defined as tissue with a hypointense signal on T1W1, an intermediate signal on T2W1 and corresponding true diffusion restriction. Masses with a score ≥ 4 are considered malignant.

**Table 2 cancers-15-05138-t002:** Descriptive characteristics of the pregnant patients.

	N	Age = m(range)
All	20	31.3(20–41)
Benign masses	13	30.3(20–37)
Malignant masses	5	32.4(26–41)
Follow-up	2	31.2(25–40)

**Table 3 cancers-15-05138-t003:** (**a**) Diagnostic performance of the expert radiologist using an SA; (**b**) diagnostic performance of the expert radiologist using the NCMS; (**c**) diagnostic performance of the expert radiologist based on both SAs and the NCMS.

(**a**)			
	Subjective Assessment		
Final Diagnosis	Benign	Malignant	Total
Benign	12	1	13
Malignant	1	6	7
Total	13	7	20
(**b**)			
	NCMS		
Final Diagnosis	Benign (<4)	Malignant (≥4)	Total
Benign	12	1	13
Malignant	1	6	7
Total	13	7	20
(**c**)			
	Value %	95% Confidence Interval	
Sensitivity	85.7	59.8–100	
Specificity	92.3	77.8–100	
Positive predictive value	85.7	59.8–100	
Negative predictive value	92.3	77.8–100	
False positive rate	7.7	0–22.2	
False negative rate	14.3	0–42.3	
Correctly classified	90.0	76.9–100	

**Table 4 cancers-15-05138-t004:** (**a**) Diagnostic performance of the non-expert radiologist using an SA; (**b**) diagnostic performance of the non-expert radiologist using the NCMS; (**c**) diagnostic performance of the non-expert radiologist based on SAs; (**d**) diagnostic performance of the non-expert radiologist based on the NCMS.

(**a**)			
	Subjective Assessment		
Final Diagnosis	Benign	Malignant	Total
Benign	6	7	13
Malignant	1	6	7
Total	7	13	20
(**b**)			
	NCMS		
Final Diagnosis	Benign (<4)	Malignant (≥4)	Total
Benign	10	3	13
Malignant	1	6	7
Total	11	9	20
(**c**)			
	Value %	95% Confidence Interval	
Sensitivity	85.7	59.8–100	
Specificity	46.2	19.1–73.3	
Positive predictive value	46.2	19.1–73.3	
Negative predictive value	85.7	59.8–100	
False positive rate	53.8	26.7–80.9	
False negative rate	14.3	0–40.2	
Correctly classified	60.0	38.5–81.5	
(**d**)			
	Value %	95% Confidence Interval	
Sensitivity	85.7	59.8–100	
Specificity	76.9	54.0–99.8	
Positive predictive value	66.7	35.9–97.5	
Negative predictive value	90.9	73.9–100	
False positive rate	23.1	0.2–46.0	
False negative rate	14.3	0–40.2	
Correctly classified	80.0	62.5–97.5	

**Table 5 cancers-15-05138-t005:** (**a**) Diagnosis outcomes with SA: inter-observer agreement between the non-expert and expert radiologists; (**b**) diagnosis outcomes with the NCMS: inter-observer agreement between the non-expert and the expert radiologists.

(**a**)				
		Non-Expert		Total
		Benign	Malignant	
Expert	Benign	7	6	13
	malignant	0	7	7
	Total	7	13	20
(**b**)				
		Non-Expert		Total
		Benign	Malignant	
Expert	Benign	11	2	13
	malignant	0	7	7
	Total	11	9	20

**Table 6 cancers-15-05138-t006:** Analytical description of the diagnoses performed by the two radiologists using the NCMS, according to each single score.

	Score 1	Score 2	Score 3	Score 4	Score 5
Expert	0	11	3	5	2
Non-expert	0	6	6	6	3

## Data Availability

The data presented in this study are available on request from the corresponding author.

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
