# Peer review of "Prediction of the Risk of Malignancy of Adnexal Masses during Pregnancy Comparing Subjective Assessment and Non-Contrast MRI Score (NCMS) in Radiologists with Different Expertise"

_cancers, 2023, doi:10.3390/cancers15215138_

Round 1
Reviewer 1 Report
The authors rise a very interesting topic. I approached the topic with great interest. The paper is written coherently and clearly. The presentation of the results is very clear. Discussion raises interesting issues. I suggest the authors consider including a bit more information about ovarian cancer in the introduction. I believe that it can significantly contribute to a better understanding of the topic. However, I leave it to the authors' decision.
Author Response
We appreciate the favourable comments of the reviewer. We have enriched the introduction since this is one of the main suggestions of also other reviewers.
Reviewer 2 Report
The manuscript " Prediction of the risk of malignancy of adnexal masses during pregnancy comparing subjective assessment and Non-Contrast MRI Score (NCMS) in radiologists with different expertise" by Panico et al. reports that the Non-Contrast MRI-based NCMS can be used for substitute MRI for characterizing ovarian mass during pregnancy. NCMS can be a helpful tool for expertise with less experience. As the first study that evaluates the NCMS in pregnant women, the authors have deeply discussed the scenarios that affect the accuracy of diagnostics. Overall, this is a methodical, well-organized manuscript that should interest readers of cancers.
Author Response
We thank very much the reviewer for the favourable comments.
Reviewer 3 Report
The authors report on the prediction of the risk of malignancy of adnexal masses during 2 pregnancy comparing subjective assessment and Non-Contrast 3 MRI Score (NCMS) in radiologists with different expertise.
A right imagistic diagnosis in such cases is fundamental for any obstetrician, so the subject is of maxim interest.
Major comments
1. The authors report that the manuscript is: “This study is the first one which evaluate the diagnostic accuracy of non-contrast MRI score in pregnant women with indeterminate ovarian masses at US.” (lines 25-26; 303-304).
Furthermore, they stated that: “There is very sparse literature on this topic, with only one study that evaluated adnexal masses in pregnant women with contrast-enhanced MRI. … Few other studies have reported data on the prevalence of malignant adnexal masses in pregnancy, but not on the diagnostic accuracy, however they have been carried out in different setting observing patients with different characteristics”.
I deeply not agree with those affirmations. There is a significant number of reports on the role of ultrasound completed with MRI, for accurate evaluate the annexal mases in pregnant women. In all reports is specified that gadolinium, although not proven teratogenic, contrast MRI is not recommended in such cases. Some of these studies have a significant number of citations, on their turn: Bethash, 2008 (!!!), 145 citations (!!!); Kvon, 2010 (!!!), 74 citations; Mancari, 2014, 24 citations; Hakoun 46 citations; Martone 2021; Cathcart, 2023; a.s.o.
So, I recommend to the authors to delete all phrases as: “This is the first…”
2. Also, referring to the previous comment, I suggest to the authors to complete the “Introduction” and “Discussion” sections with selected, important data from the literature.
Minor commnets
3. The authors report that: “The institutional Ethics committee approved this retrospective study (no. 5681) and written informed consent was obtained. Our picture archiving and communication system (PACS) was searched to retrieve MRI exams of pregnant patients scanned for an US indeterminate adnexal mass between January 2011 and February 2023.” (lines 92-95).
I invite the authors to provide the date of emission of “IRCCS no. 5681”.
4. This study included 20 patients, number which is significant for a study / manuscript. However, for an interval of 13 years – 2011-2023, this is no more too many.
So, I suggest to the authors to cooperate with other centers and to complete the group, for being significantly more interesting for the reader.
Author Response
The authors report on the prediction of the risk of malignancy of adnexal masses during 2 pregnancy comparing subjective assessment and Non-Contrast 3 MRI Score (NCMS) in radiologists with different expertise.
A right imagistic diagnosis in such cases is fundamental for any obstetrician, so the subject is of maxim interest.
Major comments
- The authors report that the manuscript is: “This study is the first one which evaluate the diagnostic accuracy of non-contrast MRI score in pregnant women with indeterminate ovarian masses at US.” (lines 25-26; 303-304).
Furthermore, they stated that: “There is very sparse literature on this topic, with only one study that evaluated adnexal masses in pregnant women with contrast-enhanced MRI. … Few other studies have reported data on the prevalence of malignant adnexal masses in pregnancy, but not on the diagnostic accuracy, however they have been carried out in different setting observing patients with different characteristics”.
I deeply not agree with those affirmations. There is a significant number of reports on the role of ultrasound completed with MRI, for accurate evaluate the annexal mases in pregnant women. In all reports is specified that gadolinium, although not proven teratogenic, contrast MRI is not recommended in such cases. Some of these studies have a significant number of citations, on their turn: Bethash, 2008 (!!!), 145 citations (!!!); Kvon, 2010 (!!!), 74 citations; Mancari, 2014, 24 citations; Hakoun 46 citations; Martone 2021; Cathcart, 2023; a.s.o.
So, I recommend to the authors to delete all phrases as: “This is the first…”
Reply
We thank the reviewer for this wise suggestion. We changed the cited phrases in the manuscript, following reviewer suggestion.
We are aware of the papers cited by the reviewer, however we have focused our attention on a) papers analysing accuracy of the MRI diagnostic imaging (only one other paper dealt with this issue using Contrast enhanced MRI); b) those papers (quite few) trying to measure prevalence of malignant masses in pregnant women indeterminate at US. Within this framework we think that the paper by Hakoun should be cited (and we have insert it in the text and references) The work by Cathcart is already reported in our paper.
- Also, referring to the previous comment, I suggest to the authors to complete the “Introduction” and “Discussion” sections with selected, important data from the literature.
Reply
We have reported the relevant suggested information both in the introduction and the discussion
Minor comments
- The authors report that: “The institutional Ethics committee approved this retrospective study (no. 5681) and written informed consent was obtained. Our picture archiving and communication system (PACS) was searched to retrieve MRI exams of pregnant patients scanned for an US indeterminate adnexal mass between January 2011 and February 2023.” (lines 92-95).
I invite the authors to provide the date of emission of “IRCCS no. 5681”
Reply.
The date of emission of ethical clearance is 05/06/2023 and now reported in the paper.
- This study included 20 patients, number which is significant for a study / manuscript. However, for an interval of 13 years – 2011-2023, this is no more too many.
So, I suggest to the authors to cooperate with other centers and to complete the group, for being significantly more interesting for the reader.
Reply.
Thanks for this suggestion. We recognize the importance of increasing the number of observations, including the organization of a multicenter study; we reported it in the paper in limitations section.
References have been updated according to the suggestions.
Reviewer 4 Report
The topic of this study is of interest, but there are many sections that have incomplete data.
- Line 176 - it was wrongly written 6 out of 30 instead of 6 out of 20
- The study by Bird et al highlighted the MRI examination rate with contrast at 0.12% (n=860) of all pregnancies, 22.3% being abdominal and pelvic, 70.2% of exposures being in the first trimester. DOI: 10.1148/radiol.2019190563. The risk is especially high in the first trimester due to the large number of MRI examinations according to the study above.
- Normally, the scanning time with a contrast agent (gadolinium) is shorter than MRI without a contrast agent. In the case of suspicion of oncological diseases, it is preferable to perform an MRI examination with contrast in carefully selected cases.
- What was the distribution by trimesters of adnexal masses in pregnancy?
- The MRI sequences were acquired on a 1.5 T. How do you think the accuracy of the results on a 3 T MRI can be influenced? The American College of Radiology has affirmed the safety of MRI with powers up to 3 T, without adverse effects on the embryo or fetus and organogenesis.
- What were the imaging characteristics in the MRI examinations (bilaterality, dimensions of the mass, type of lesion)?
- In how many asymptomatic patients were incidentally detected adnexal masses?
- Discussions should include the position of the Royal College of Radiologists in Great Britain, which recommended that GBCAs should only be used when essential diagnostic information cannot be obtained with unenhanced scans.
- Demographic data of the group under study, of the distribution of the detection of adnexal masses by trimesters, of the sizes (NMR without contrast usually detects large formations), are not present, and there are no data regarding the histopathological result in the conditions where in the design of the study it was mentioned this thing.
- Where is the result of the imaging evaluation of the cases mentioned at one year? (This was mentioned in the study design)
- Not all cases are presented at the end of the article. It would also be interesting to evaluate the other cases, especially those with oncological conditions.
Kind regards
Author Response
The topic of this study is of interest, but there are many sections that have incomplete data.
- Line 176 - it was wrongly written 6 out of 30 instead of 6 out of 20
Reply.
Thank you for this suggestion. We have proceeded with the correction.
- The study by Bird et al highlighted the MRI examination rate with contrast at 0.12% (n=860) of all pregnancies, 22.3% being abdominal and pelvic, 70.2% of exposures being in the first trimester. DOI: 10.1148/radiol.2019190563. The risk is especially high in the first trimester due to the large number of MRI examinations according to the study above.
Reply.
Thank you for this suggestion. We have inserted this information in the paper.
Normally, the scanning time with a contrast agent (gadolinium) is shorter than MRI without a contrast agent. In the case of suspicion of oncological diseases, it is preferable to perform an MRI examination with contrast in carefully selected cases.
Reply.
Even if we agree with the reviewer, contrast may be useful in extremely selected cases and only if necessary (very few cases), also considering that, in our country, national scientific societies discourage the use of contrast in pregnant women.
- What was the distribution by trimesters of adnexal masses in pregnancy?
Reply
Unfortunately, we cannot have this information for all patients (some of them are old cases and not all of them were clinically followed in our center); because of that we decided not to insert this piece of information
- The MRI sequences were acquired on a 1.5 T. How do you think the accuracy of the results on a 3 T MRI can be influenced? The American College of Radiology has affirmed the safety of MRI with powers up to 3 T, without adverse effects on the embryo or fetus and organogenesis.
Reply.
3T MRI is safe, but we think that the added value can be reduced by possible motion artefacts from the foetus or the mum, considering the discomfort that can be present in MRI.
- What were the imaging characteristics in the MRI examinations (bilaterality, dimensions of the mass, type of lesion)?
Reply.
We have decided to describe the most interesting cases in details. The other cases show the most common MRI features and this is why we decided not going in details for them
- In how many asymptomatic patients were incidentally detected adnexal masses?
Reply: All patients but two were asymptomatic; the two had bloating symptoms correlated with carcinomatosis.
- Discussions should include the position of the Royal College of Radiologists in Great Britain, which recommended that GBCAs should only be used when essential diagnostic information cannot be obtained with unenhanced scans.
Reply.
We have reported the position of ESUR (European Society of Urogenital Radiology) which takes into account also the national positions. This paper recommends the use of enhanced MRI in case the diagnosis is not clear.
- Demographic data of the group under study, of the distribution of the detection of adnexal masses by trimesters, of the sizes (NMR without contrast usually detects large formations), are not present, and there are no data regarding the histopathological result in the conditions where in the design of the study it was mentioned this thing.
Reply.
Information on demographic data have been added in Supplementary table 1 with information about each patient
- Where is the result of the imaging evaluation of the cases mentioned at one year? (This was mentioned in the study design)
Reply.
Two patients had follow up because of benign aspect of ovarian masses; now this information is written in the result section (table 2)
- Not all cases are presented at the end of the article. It would also be interesting to evaluate the other cases, especially those with oncological conditions.
Reply.
As pointed out before we have decided to describe the most interesting cases in detail, also considering the rules of the journal that limit the number of words and figures.
References have been updated according to the suggestions
Round 2
Reviewer 3 Report
The authors significantly improved the manuscript.
But they still have to clarify the etical aspect regarding the beginning start of the study - 2011 and: "The date of emission of ethical clearance is 05/06/2023..."
Author Response
Thanks to the reviewer for the comment, as Ethical issues are important.
This is a retrospective study: we had ethical approval and started the research only after that. We modified the section "Study design" to clarify it, and we added the date of approval in the text.
Reviewer 4 Report
The authors made important improvements and clarifications to the article.
Kind regards
Author Response
We want to thank the reviewer for the valuable suggestion.
Round 3
Reviewer 3 Report
The authors clarified my concern.
Author Response
Thank you for your comment